# Effect of Silviculture on Carbon Pools during Development of a Ponderosa Pine Plantation

**Jie Zhang [1], Jianwei Zhang [2],\* , Kim Mattson [3] and Kaelyn Finley [2]**

[1] College of Forestry, Northeast Forestry University, Harbin 150040, China; zj_1212@nefu.edu.cn
[2] Pacific Southwest Research Station, USDA Forest Service, 3644 Avtech Parkway, Redding, CA 96002, USA; kaelyn.a.finley@usda.gov
[3] Ecosystems Northwest, 189 Shasta Avenue, Mount Shasta, CA 96067, USA; mattson@ecosystemsnw.com
\* Correspondence: jianwei.zhang2@usda.gov

**Abstract:** Forest stands can be considered as dynamic carbon pools throughout their developmental stages. Silvicultural thinning and initial planting densities for reforestation not only manipulate the structure or composition of vegetation, but also disturb forest floor and soils, which, in turn, influences the dynamics of carbon pools. Understanding these carbon pools both spatially and temporally can provide useful information for land managers to achieve their management goals. Here, we estimated five major carbon pools in experimental ponderosa pine (*Pinus ponderosa*) plots that were planted to three levels of spacing and where competing vegetation was either controlled (VC) or not controlled (NVC). The objectives were to determine how an early competing vegetation control influences the long-term carbon dynamics and how stand density affects the maximum carbon (C) sequestration for these plantations. We found that planting density did not affect total ecosystem C at either sampling age 28 or 54. Because of competing vegetation ingrowth, the NVC ($85 \pm 14$ Mg ha$^{-1}$) accumulated greater C than the VC ($61 \pm 6$ Mg ha$^{-1}$) at age 28. By age 54, the differences between treatments narrow with the NVC ($114 \pm 11$ Mg ha$^{-1}$) and the VC ($106 \pm 11$ Mg ha$^{-1}$) as the pines continue to grow relatively faster in the VC when compared to NVC and C of ingrowth vegetation decreased in NVC, presumably due to shading by the overstory pines. The detritus was not significantly different among treatments in either years, although the mean forest floor and soil C was slightly greater in NVC. While NVC appears to sequester more C early on, the differences from the VC were rather subtle. Clearly, as the stands continue to grow, the C of the larger pines of the VC may overtake the total C of the NVC. We conclude that, to manage forests for carbon, we must pay more attention to promoting growth of overstory trees by controlling competing vegetation early, which will provide more opportunities for foresters to create resilient forests to disturbances and store C longer in a changing climate.

**Keywords:** carbon pools; carbon sequestration; challenge experimental forest; competing vegetation control; long-term forest experiment; *Pinus ponderosa*; stand density

## 1. Introduction

Forests store about 860 petagrams of terrestrial carbon (C) and sequester about one-third of the annual anthropogenic fossil fuel C emissions from the atmosphere [1]. However, biotic and abiotic disturbances that are intensified by climate change, such as wildfires and bark beetle outbreaks [2], have not only released carbon into the atmosphere, but also reduced the capacity for C sequestration in these forests. Therefore, appropriate silvicultural techniques for improving the health of forest stands and enhancing forest resiliency are necessary for managing forests for carbon.

Ponderosa pine (*Pinus ponderosa* Lawson & C. Lawson) is a common species grown in both natural forests and plantations for timber production in the western forests of North America. The drought tolerance and fire resistance of this species make it a favorable species for rapid reforestation after stand-replacing wildfires as well as following commercial harvests. In California, about 162,000 ha in National Forest land and 128,000 ha of private (forest industry) lands [3] are covered by ponderosa pine plantations. These plantations have also been managed for carbon sequestration in mitigating global climate warming because of its relatively high growth rate and occurrence across large areas [4].

Forest development is dynamic and different stages are associated with specific strategies for achieving management goals. For example, after a stand-replacing wildfire in the ponderosa pine ecosystem, the fastest and more effective means for reforestation is to plant trees, because natural regeneration is not dependable for two reasons. One is a lack of seed sources after a large and high severity fire, as seeds cannot travel long distance from adjacent non-burned forests [5]. Another is shrub competition hindering seedling survival and growth should seeds germinate [6,7]. Major shrubs in ponderosa pine dominant forests, such as *Arctostaphylos* and *Ceanothus* spp., with abundant seed banks, quickly germinate after the wildfires or the post-harvest site, and grow aggressively, outcompeting the regenerated tree species for resources. For example, on a productive site in the Sierra Nevada of California, 20-year-old pine stands with dense shrubs achieved about half the volume of brush-free stands [8]. At a poor site on the foothill of Mount Shasta, McDonald and Powers [9] showed that 30-year-old ponderosa pine trees growing with dense shrub cover produced less than one percent of the volume of trees growing free of shrub competition. Therefore, competition control in the early stage of plantations is essential until it can close its canopy [7,10]. At that point, an onset of inter-tree competition often occurs, which requires density management to improve the health and growth of the pine plantations as inter-tree competition could be more intensive than pine-shrub competition [7]. Across various growing stock levels in the Sierra Nevada, Zhang et al. [11] found that thinning an existing 20-year-old plantation to less than half of the amount of basal area would not reduce the total biomass production at year 60, but would position the stand for increased pine growth in the future years.

Although the effects of competing vegetation control and density management on stand growth have demonstrated enhanced pine growth in ponderosa pine plantations [6,7,12], studies of the effects on C allocation and storage in different pools during stand development are rare. Most of the studies have used chronological estimations that were conducted with different stands of varying ages and ignoring the site-specific interactions [13–15]. Here, we use measurements from the same site 26 years apart. A better understanding of the patterns of C accumulation as the stands age and the effects of silviculture practices will aid in determining management strategies that maximize carbon sequestration, while also maintaining productive capacity and provisioning of other ecosystem services.

In this study, we measured the C stocks among different pools for a ponderosa pine plantation. We compared the results associated with management treatments consisting of three planting densities, with and without competing vegetation controls. We compared measurements made at age 28 to those at age 54. The major goals were to determine how early competing vegetation control influences the C dynamics and what stand density provides the maximum C sequestration over the first 50 years of development.

## 2. Materials and Methods

### 2.1. Study Site and Original Design

Study plots are at the Challenge Experimental Forest, located on the west slope of the northern Sierra Nevada, the Plumas National Forest in Yuba County, California (lat. 39.4756 N., long. 121.2167 W., Elev. 813 m). The climate is Mediterranean with warm and dry summers and cool and wet winter (Figure 1). The mean annual precipitation averages 1727 mm, 98 percent of which falls between October and May. The annual temperatures normally range from 6 °C in January to 21 °C in July.

The combination of high precipitation and mild temperature produces exceptionally favorable growing conditions. The soil is deep and derived from metamorphosed basalt (mesic Xeric Haplohumults). The site index is high (34 m at age 100).

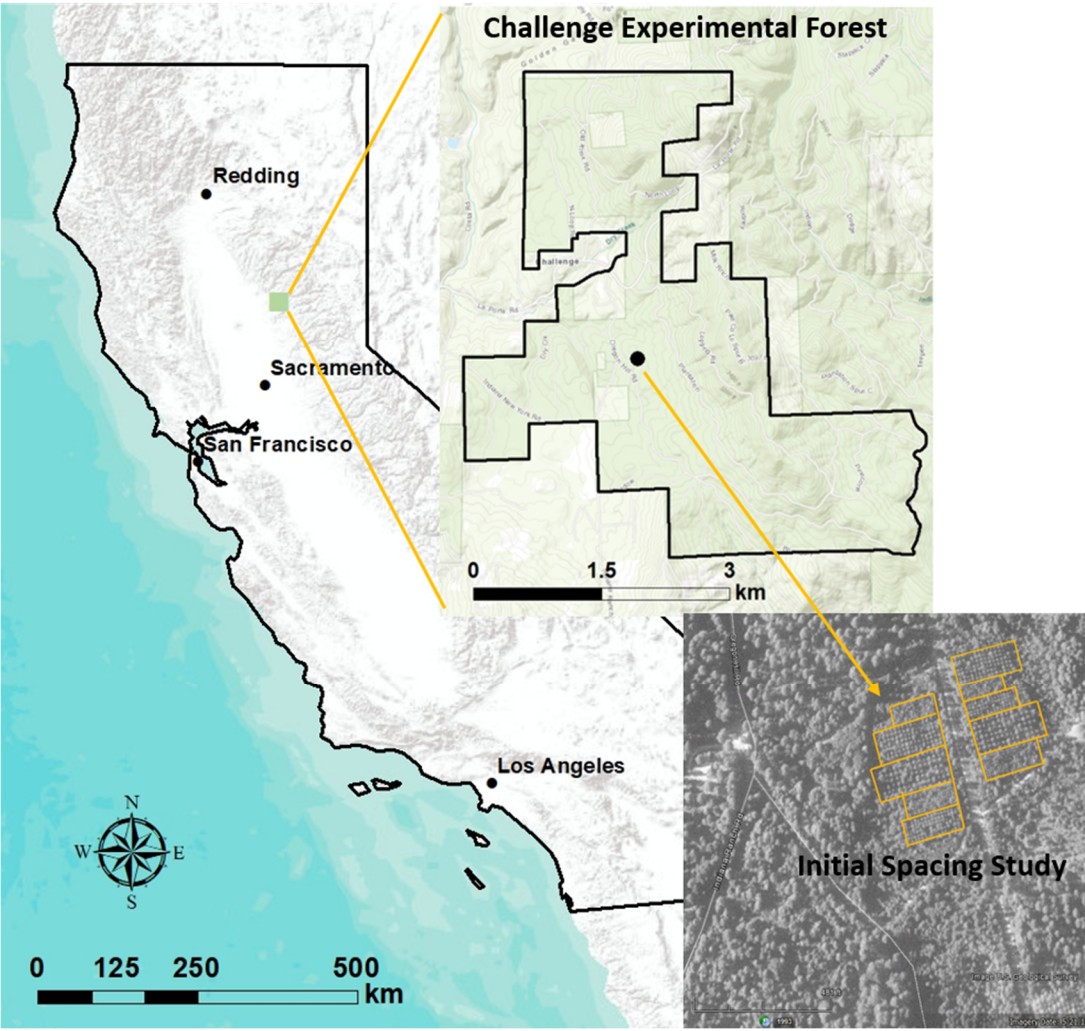

**Figure 1.** Map of California, Challenge Experimental Forest, and Initial Spacing Study plot layout.

The experimental plots were established with planted ponderosa pine seedlings in 1966 [8]. The previous stand was composed of 70-year-old ponderosa pine that was clearcut for this experiment. Logging slash was raked, piled, and burned off the unit, leaving little remaining forest floor on site. There are two blocks; each block contains five plots that were randomized to receive one of five levels of planted pines at square spacings of 1.8, 2.7, 3.7, 4.6, and 5.5 m. Each plot was split into two adjacent subplots with four rows for 1.8 m, three rows for 2.7 m, and two rows for other spacings, respectively, of buffer trees that were planted at the same spacing as the measurement trees. On one subplot, competing vegetation was controlled by 2,4,5-T (2,4,5-trichlorophenoxy acetic acid) that was applied by hand sprayer in the second and fourth years after planting. Subsequent competing woody plants were removed by hand for about five more years (VC). The herbicide spray never touched the planted seedlings and, therefore, should not affect their performance. On the other subplot, all planted pines plus competing vegetation ingrowth were allowed to develop naturally with no vegetation control (NVC). In the current analysis, we only focus on two measurements (ages 28 and 54) for three of the spacing treatments—the narrowest (1.8 m), intermediate (3.7 m), and widest (5.5 m), because only

these plots were also measured for C pools at age 28. Table 1 shows the general stand characteristics for these subplots.

**Table 1.** Stand characteristics of ponderosa pine subplots used in this study at Challenge Experimental Forest, California. Treatments are three levels of planted spacings of pine and two vegetation controls of competing vegetation in a split plot design.

| Competing Vegetation Treatment | Stand Age | Overstory Planting Spacings | TPH [1] | Height (m) | QMD [2] (cm) | BA [3] (m² ha⁻¹) |
|---|---|---|---|---|---|---|
| **Vegetation Control** | Age 28 | 1.8 × 1.8 m | 2621 | 10.2 | 13.7 | 39 |
| | | 3.7 × 3.7 m | 747 | 14.2 | 24.6 | 35.6 |
| | | 5.5 × 5.5 m | 318 | 15.2 | 32.2 | 25.8 |
| | Age 54 | 1.8 × 1.8 m | 1248 | 19.9 | 20.9 | 42.4 |
| | | 3.7 × 3.7 m | 622 | 23.7 | 32.6 | 52.2 |
| | | 5.5 × 5.5 m | 263 | 26.8 | 45.8 | 43.4 |
| **No vegetation Control** | Age 28 | 1.8 × 1.8 m | 2746 | 9.7 | 12.6 | 34.3 |
| | | 3.7 × 3.7 m | 747 | 11.3 | 20.2 | 24.4 |
| | | 5.5 × 5.5 m | 332 | 13.3 | 26 | 17.6 |
| | Age 54 | 1.8 × 1.8 m | 1123 | 19.5 | 21.8 | 43.7 |
| | | 3.7 × 3.7 m | 622 | 19.9 | 30.5 | 46.7 |
| | | 5.5 × 5.5 m | 318 | 23.1 | 39.1 | 38.2 |

[1] TPH = trees per hectare; [2] QMD = quadratic mean diameter; [3] BA = basal area.

*2.2. Aboveground Carbon for Planted Ponderosa Pine*

Height and DBH (diameter at 1.37 m height from the ground) measurements had been frequently conducted on the planted ponderosa pine [7,8,16], including at ages 28 and 54 when other C pools were quantified.

Individual tree biomass was estimated while using allometric equations that were established by harvesting 30 trees over a range of DBH from trees in the buffer zones of the 1.8, 3.7, and 5.5 m spacing plots in VC and NVC at age 28. The disks were removed from the bole at the stump (0.3 m), breast height (1.37 m), and the middle of the live crown for each felled tree. The dimensions (diameter of two axes inside and outside bark and disk thickness at the end of each axis) of each sampled disk and length of each bole section were recorded in the field.

For each of the felled trees, the live crown was divided into five equal sections and all living branches were measured for diameter at a standard distance from the bole of tree. The branch diameters were then squared, summed, and averaged in the field. The top of the main stem, as well as the branch of average squared diameter from each sector, were collected for later weighing the wood and foliage separately. Below the live crown, the dead branches were sampled in a similar way.

The samples for bole wood, branches, and foliage were all weighed separately after drying to constant weight at 80 °C. The total bole biomass was estimated for each section as the product of the average of density of the ends of each section and the volume of that section. Separate allometric equations were established for the different planting density and vegetation control treatments from the branch and foliage measurements; these equations were used in order to estimate all branches and associated foliage on the tree stem. The total aboveground biomass for an individual tree was the sum of all components (foliage, branch, and bole).

We used the pooled data to obtain the following allometric equation because the specific fits for the various treatments of planting densities by vegetation control were not significantly different.

$$AGB = 0.0819\ DBH^{2.3898}, \text{r}^2 = 0.98 \tag{1}$$

where *AGB* is aboveground biomass in kg and *DBH* is in cm. We estimated total aboveground biomass for each tree within the subplots at ages 28 and 54 years. The total subplot biomass was a summation of all trees in the subplot and using a conversion ratio of 50% biomass to C. Subsequently,

subplot-level aboveground C (Mg ha$^{-1}$) was calculated. The same procedures were used for the dead tree aboveground biomass as mortality C. Live root C was assumed that root/shoot equals 0.22 for planted trees overall, which were found in another study (Jianwei Zhang, unpublished data) and it was not greatly different from the mean of 0.26 for gymnosperms from the literature [17].

### 2.3. Competing Vegetation Carbon

The competing vegetation included all naturally regenerated plant species, i.e., all trees and shrubs, excluding the planted ponderosa pine. At age 28, all of the competing vegetation was collected at five 0.785 m$^2$ circular plots randomly located on each subplot. The competing vegetation was separated into three categories: naturally regenerated tree seedlings, shrubs, and herbaceous plants and brought to the laboratory for dry weights after drying at 80 °C to constant weight.

At age 54, within each subplot, competing vegetation consisting of shrubs below <1.5 m in height and herbaceous vegetation was sampled in five, 0.25 m$^2$ square subplots. The aboveground live portions were weighed in the field; and, for those samples >than 300 g, at least 300 g subsamples dried at 80 °C for 48 h for moist-to-dry-weight conversions. Natural ingrowth of tree and larger shrub biomass was estimated from DBH while using allometric equations for the specific species: *Abies concolor*, *Calocedrus decurrens,* and *Pinus lambertiana* (Jianwei Zhang, unpublished data), *Arbutus menziesii* and *Notholithocarpus densiflorus* [18], *Arctostaphylos patula* [19], and *Quercus kelloggii* [20]. The subplot-level aboveground C (Mg ha$^{-1}$) of ingrowth vegetation was calculated while using similar approach as for the planted pines except that the live root C was assumed to have a root/shoot ratio of 0.25 (Jianwei Zhang, unpublished data), which was the same mean for angiosperms that were reviewed by Cairns et al. [17].

### 2.4. Forest Floor and Soil Carbon

In both sampling years, the forest floor samples were collected at five random locations per subplot using a 0.25-m$^2$ frame. The samples were weighed moist in the field and, as with understory, a subsample was collected for moist-to-dry-mass conversions and for carbon stock estimations. At age 28, woody debris was included in the forest floor samples due to the small quantity. At age 54, woody debris was sampled for volume for diameters >2.5 cm while using 20-m long transects [21,22]. Volumes were converted to mass by assigning mean density by decay classes assessed in the field. Down dead pine boles were not counted as wood input because they were kept as part of the vegetation sampling and classed as dead pines. Mineral soil cores were taken from the 0–5, 5–10, and 10–20 cm depths at five spots in each subplot at age 28 using a soil auger. Soil bulk density was determined from these samples by dividing soil sample volume by total soil sample weight after drying at 105 °C for 48 h. The five samples per subplot were then composited by depth for each subplot. Soil organic C concentration at each depth was analyzed by dry combustion at 1300 °C [23] and C stocks, were calculated as the product of the mean soil bulk density value and carbon concentration. At stand age of 28, soil C was converted to the soil depths of 0–10 cm and 10–20 cm, so as to match the latest inventory. At age 54, mineral soil samples from 0–10 and 10–20 cm at 3 locations per subplot were collected while using a hammer-driven soil bulk density sampler. The soil was collected inside removable sleeves contained in the soil corer. Once the sleeves were removed with the collected soil, the top and bottom were carefully trimmed away to obtain a more exact volume of soil. The samples were assayed for total bulk density individually and the three samples were composited by depth for each subplot for soil organic C by Oregon State University soils laboratory while using the same methods as at age 28.

### 2.5. Statistical Analysis

We analyzed all of the variables based on a split-plot, randomized, complete block design. Tree density was the main plot effect and understory vegetation (VC and NVC) the subplot effect.

All were set as fixed effects while block was set as a random effect in SAS PROC MIXED (SAS Institute Inc. 2012). The full statistical model is:

$$y_{ijk} = \mu + \alpha_i + \varepsilon_{1ik} + \beta_j + \alpha\beta_{ij} + \gamma_k + \varepsilon_{ijk} \tag{2}$$

where $y_{ijk}$ is the dependent variable summarized for the $i$th tree density, $j$th shrub cover, and the $k$th block, $\mu$ is the overall mean, $\alpha_i$ and $\beta_j$ are the fixed effects of the $i$th tree density ($i$ = 1, 2, . . . , and 5) or $j$th shrub cover ($j$ = 1 and 2), $\gamma_k$ is the random effect of the $k$th block ($k$ = 1 and 2), $\gamma_k \sim N\left(0, \sigma_B^2\right)$, and $\varepsilon_{1ik}$ is an experimental error to test the main plot effect, $\varepsilon_{ijk}$ is an experimental error for testing the subplot effect, and the rest of the terms, $\varepsilon_{1ik} \sim iid\ N\left(0, \sigma_{e1}^2\right)$, and $\varepsilon_{ijk} \sim iid\ N\left(0, \sigma_e^2\right)$.

For each variable, residuals were examined to ensure that statistical assumptions of normality and homoscedasticity were met. If not, a natural log transformation was applied. During the model selection process, we selected the model with the minimum Akaike Information Criterion (AIC). Multiple comparisons among treatments were conducted for least squares means while using the Tukey–Kramer test by controlling for the overall $\alpha$ = 0.05.

## 3. Results

### 3.1. 28-Year-Old Stand Carbon

#### 3.1.1. Aboveground Vegetation Carbon

Carbon (C) mass of the planted ponderosa pines was significantly greater when competing vegetation was controlled as compared to when it was not controlled (means of 59.1 vs. 42.2 Mg C ha$^{-1}$; $P$ < 0.04, Table S1, Figure 2a). There was very little competing vegetation C in the VC plots and considerable amounts in the NVC plots (Figure 2b). The competing vegetation C, mostly *Arctostaphylos patula*, in the NVC plots was the same magnitude as the C of the planted ponderosa pines and it more than made up for the reduction in C of the planted ponderosa pine C in all spacings. The result was greater total aboveground C including both the planted trees and competing vegetation in the NVC plots at year 28 than the VC plots (85.0 vs. 61.0 Mg C ha$^{-1}$). Competing vegetation C further increased as the planted pine spacing increased, demonstrating the shading effects of denser plantings of pine on the growth of the competing vegetation.

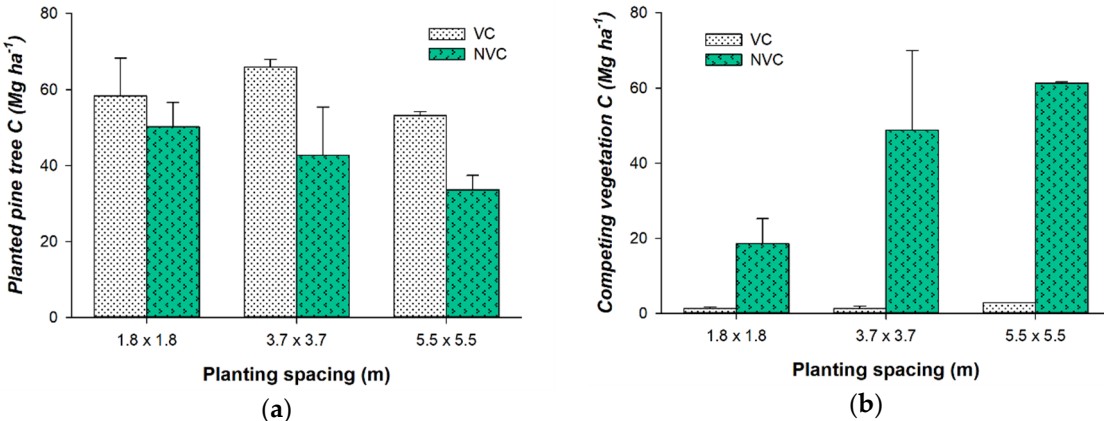

(a)                                                        (b)

**Figure 2.** 28-year-old aboveground vegetation carbon (mean + sd) for (**a**) planted trees and (**b**) competing vegetation grown under overstory planted at three levels of spacing and competing vegetation control (VC) and non-control (NVC) at the Challenge Experimental Forest, California.

Although no statistical difference was found in total aboveground vegetation C among spacings, there was a tendency that 3.7-m spacing produced the most total C for both VC and NVC treatments.

The two plots that produced the least C were the NVC 1.8-m spacing and in the VC 5.5-m spacing. The NVC 1.8-m spacing may have acted to shade the competing vegetation, and the fewer trees in the VC 5.5-m spacing did not produce proportionately more growth.

### 3.1.2. Forest Floor Carbon

At age 28, the forest floor accumulated substantial amounts of C with significantly ($p < 0.04$) more in NVC than in VC (means 21.2 vs. 15.5 Mg C ha$^{-1}$; Table 2 and Table S1). The greatest forest floor accumulations in the NVC plots were those within the wider pine density spacings, which had also had the most understory growth accumulations (Figure 2b) suggesting the forest floor accumulation was driven mostly by the competing vegetation. In contrast, for the vegetation controlled plots (VC), the highest density plots accumulated the most forest floor C showing the effect of greater pine needle fall in the higher density plots. The forest floor C accumulation was mostly driven by the O horizon C as no treatment effects were found for woody debris and no other terms were statistically significant.

**Table 2.** 28-year-old ponderosa pine plantation detrital carbon pools (Mg C ha$^{-1}$) at Challenge Experimental Forest, California. Treatments are three levels of planted spacings of pine and two vegetation controls of understory in a split plot design.

| Understory Treatment | Layer [1] | Overstory Planting Spacings | | |
|---|---|---|---|---|
| | | 1.8 × 1.8 m | 3.7 × 3.7 m | 5.5 × 5.5 m |
| | Woody debris | 1.4 | 0.4 | 0.3 |
| | O horizon | 16.7 | 13.7 | 13.8 |
| | **FF Total** | **18.1** | **14.1** | **14.2** |
| **Vegetation control** | Soil (0–10 cm) | 32.5 | 42.1 | 38 |
| | Soil (10–20 cm) | 40.4 | 49.1 | 51.2 |
| | **Soil Total** | **72.9** | **91.2** | **89.2** |
| | *Grand Total* | *91.1* | *105.3* | *103.2* |
| | Woody debris | 1 | 1.3 | 0.6 |
| | O horizon | 18.1 | 19.4 | 23.4 |
| | **FF Total** | **19.1** | **20.7** | **23.7** |
| **No vegetation control** | Soil (0–10 cm) | 38.8 | 42.3 | 36.9 |
| | Soil (10–20 cm) | 53.4 | 54.3 | 31.1 |
| | **Soil Total** | **92.2** | **96.6** | **68** |
| | *Grand Total* | *111.3* | *117.3* | *91.7* |

[1] Woody debris = pieces > 2.5 cm diameter; FF = forest floor.

### 3.1.3. Soil Carbon

Soil C was somewhat variable among the plots, but it showed no significant treatment effects, with a general trend that the deeper soil at 10–20 cm soil stored more C than the top layer of soil, with the exception of the NVC 5.5-m spacing (Table 2). This was due to its greater bulk density at depth. The planting spacing effect on soil C was opposite the trend as the forest floor, which is, treatments with the largest forest floor C had the smallest soil C mass.

### 3.1.4. Summary at Age 28

Total ecosystem C was greater in NVC ($p < 0.05$) by 19–23% (Figure 3). Within the vegetation control treatments, no significant difference was detected among the planted pine spacings although total C was the greatest in 3.7 × 3.7 m spacing in both vegetation treatments (and the least in 1.8 × 1.8 m spacing. Across a combination of treatments, the top 20-cm soil carbon accounted over 40%; planted pine trees were about 40% in VC plots, but about 25% in NVC plots. As expected, competing vegetation

was clearly substantial in the NVC plots, accounting for an increasing 12%, 28%, and 38% of total C as the spacing increased.

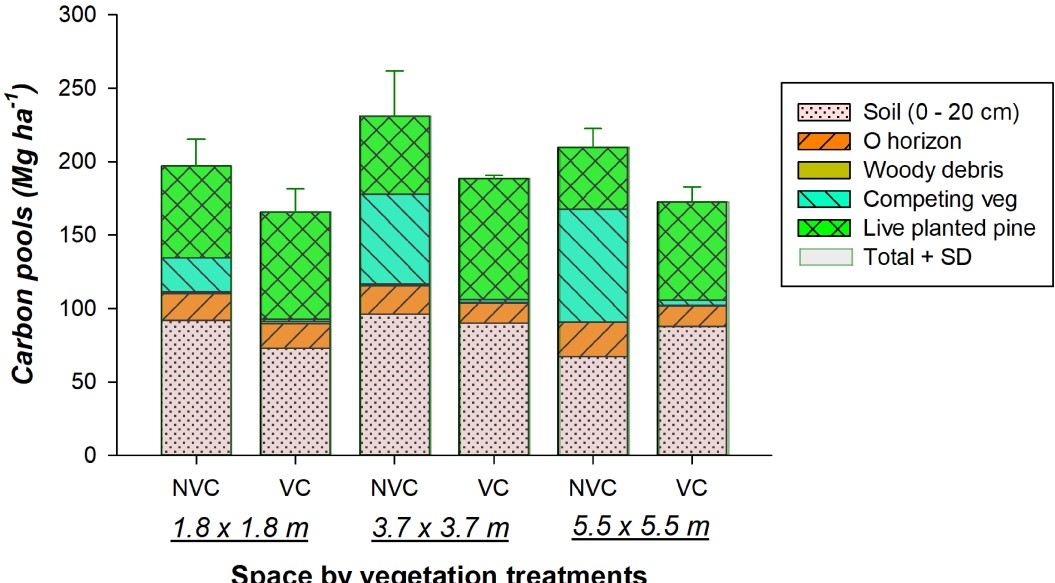

**Figure 3.** Age 28 total ecosystem carbon pools in ponderosa pine plantations under combinations of understory vegetation control (VC) and non-control (NVC) and three different levels of planting spacings at Challenge Experimental Forest, California, USA.

### 3.2. 54-Year-Old Stand Carbon

#### 3.2.1. Aboveground Vegetation Carbon

At 54 years, C patterns had shifted. The VC treatments had nearly the same C as the NVC in aboveground vegetation (means 106 vs. 114 Mg C ha$^{-1}$). The effects of planting spacing also had largely disappeared and mortality was showing up (Figure 4, Table S1). Total planted tree C showed a larger variation in NVC than in VC, as indicated in the larger standard errors. The high density plots (1.8 m) experienced a heavier stand self-thinning.

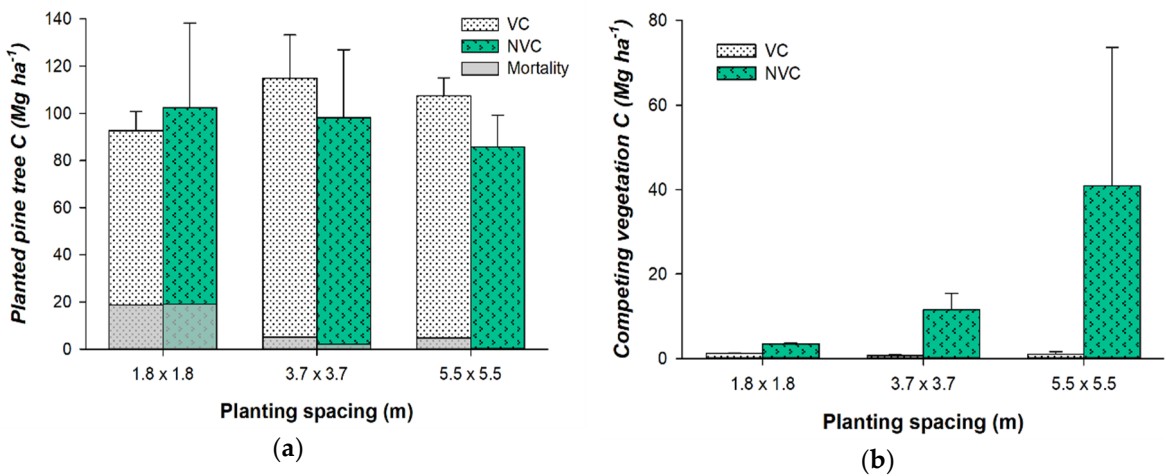

**Figure 4.** 54-year-old aboveground vegetation carbon (mean + sd) for (**a**) planted pine trees with mortality (shaded area) and (**b**) competing vegetation grown under three levels of pine spacing and two competing vegetation control treatments at Challenge Experimental Forest, California, USA.

During the 26 years between measurements, the planted pine had nearly doubled in C with a small amount of mortality. At the same time, competing vegetation C had been reduced by approximately half. The understory C had also transitioned from mostly shrubs at age 28, to more herbaceous plants, as well as both hardwood and conifer ingrowth at age 54.

A significant difference in competing vegetation C was still present between NVC and VC ($p < 0.01$). There was a significant interaction among spacing by vegetation control ($p = 0.015$, Table S1). The NVC plots still showed substantially more understory vegetation C in the more open planted plots which primarily resulted from a considerable increase in hardwood tree ingrowth.

### 3.2.2. Forest Floor Carbon

Forest floor C patterns also shifted by age 54. The NVC had essentially no change in forest floor C, suggesting these forest floors with large deciduous inputs had reached a sort of steady state. The VC showed substantial increases in forest floor C at age 54 in the two more densely spaced pines treatments, which suggested the trend observed at age 28 of greater needle inputs in the denser stands was still occurring and causing forest floor mass increase (Table 3). The forest floor C in the plots with the most widely spaced pines of VC was essentially unchanged or perhaps slightly smaller, likely the result of its smaller litterfall inputs. By age 54, forest floor C did not significantly differ between VC and NVC ($p > 0.46$, Table S1). The spacing effect and interactions between spacing and vegetation control were close to the significant levels ($p = 0.094$ and 0.065). There were still very low amounts of woody debris that accumulated in these plots, mainly because all down woody debris from the pine tree mortality was assigned into the pool of planted trees and not the pool of dead wood.

**Table 3.** Carbon pools (Mg C ha$^{-1}$) in forest floor and soil for 54-year-old plantation at Challenge Experimental Forest, California, USA. Treatments are three levels of planted spacings of pine and two vegetation controls of understory in a split plot design.

| Understory Treatment | Layer [1] | Overstory Planting Spacings | | |
|---|---|---|---|---|
| | | **1.8 × 1.8 m** | **3.7 × 3.7 m** | **5.5 × 5.5 m** |
| **Vegetation control** | Woody debris | 0.7 | 0.5 | 0.2 |
| | O horizon | 32.4 | 23.3 | 10.1 |
| | **FF Total** | **33.1** | **23.8** | **10.3** |
| | Soil (0–10 cm) | 45.8 | 44.6 | 58.2 |
| | Soil (10–20 cm) | 32.9 | 28.9 | 31.6 |
| | **Soil Total** | **78.7** | **73.5** | **89.8** |
| | *Grand Total* | *111.8* | *97.3* | *100.1* |
| **No vegetation control** | Woody debris | 0.9 | 1.2 | 1.6 |
| | O Horizon | 19.8 | 23.6 | 18.4 |
| | **FF Total** | **20.7** | **24.8** | **20** |
| | Soil (0–10 cm) | 55.7 | 45.3 | 49.6 |
| | Soil (10–20 cm) | 41.4 | 36.4 | 38 |
| | **Soil Total** | **97.1** | **81.7** | **87.6** |
| | *Grand Total* | *117.8* | *106.5* | *107.6* |

[1] Woody debris = pieces > 2.5 cm diameter; FF = forest floor.

### 3.2.3. Soil Carbon

Total soil C mass estimates were relatively similar between ages 28 and 54, although perhaps less variable by age 54 (Table 3). At age 28, the mean of total soil C for the vegetation control treatments were essentially the same or only slightly greater in NVC when averaged across the different planting spacings. By age 54, the NVC plot means were significantly greater than VC plot means ($p < 0.03$; Table S2) with +18.4 and +8.2, Mg ha$^{-1}$ at 1.8 and 3.7 m spacings, respectively, although there was slight

decrease of −2.2 Mg ha$^{-1}$ at 5.5 m space. The top 10-cm soil C was greater than the 10–20 cm depth ($p < 0.01$), with 49.9 Mg ha$^{-1}$ at top 10 cm and 34.9 Mg ha$^{-1}$ at the 10–20 cm. No significant difference in soil C was detected among the different planting spacings and neither were any interactions (i.e., a combination of spacing, vegetation control, and depth; $p > 0.11$).

### 3.2.4. Summary at Age 54

In contrast to age 28, there were no longer significant differences in total ecosystem carbon ($p > 0.36$; Table S1), not only between VC and NVC, but also among different spacings as well as their interactions. The total C had increased over the 26 intervening years by about 50 Mg C ha$^{-1}$ and this was mostly in the planted pines. At age 28, C by treatments ranged from 150 to 230 Mg C ha$^{-1}$ and, by age 54, it ranged from 226–263 Mg C ha$^{-1}$ (Figure 5). There was still a remaining trend of greater total C in NVC versus VC. This was because of the trends of slightly greater C in NVC in most of the pools measures. This difference was considerably smaller as pine trees sequestered increasing amounts of C in the VC compared to the NVC. Soil C accounted for over 35% at the top 20 cm (Figure 5); planted ponderosa pine trees were about 50%, including 4% dead trees, which primarily occurred in the 1.8 m spacing due to higher competition. Understory vegetation clearly declined in the NVC plots in comparison to age 28, and it was only substantial in wider spacings, consisting mainly of hardwood species *A. menziesii*, *N. densiflorus*, and *Q. kelloggii*. At 3.7 and 5.5 m spacing, understory vegetation accounted 6% and 19% of total ecosystem C, respectively. As expected, mortality was substantial in the 1.8 m spacing plots as self-thinning onset after age 28.

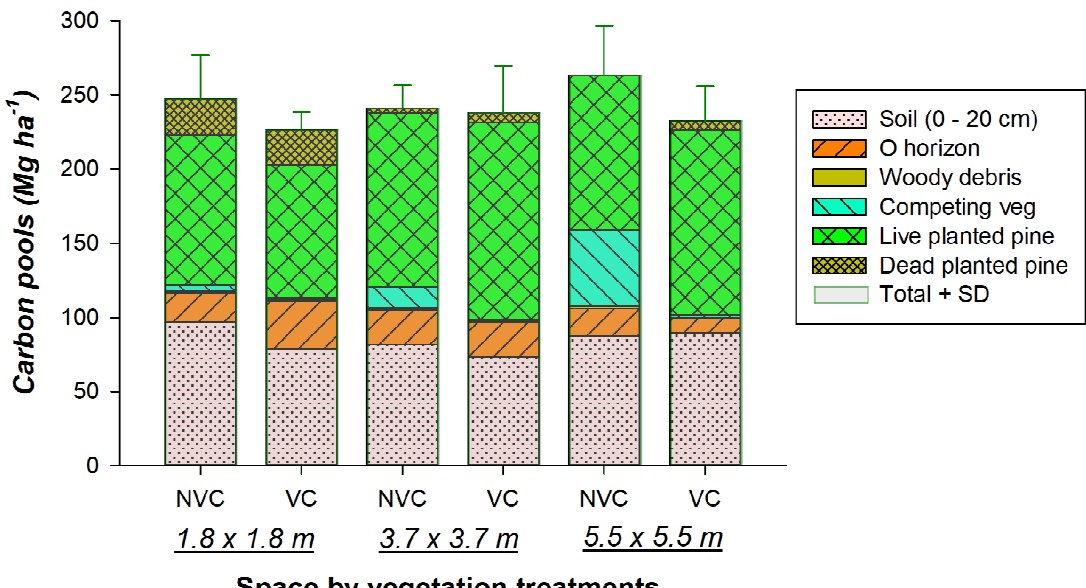

**Figure 5.** Age 54 total ecosystem carbon among pools in ponderosa pine plantations (as in Figure 3) under combinations of understory vegetation control (VC) and non-control (NVC) and three different levels of planting spacings at Challenge Experimental Forest, California, USA.

## 4. Discussion

The results from this study demonstrate the gradual C build-up during stand development. Forest stand C pools shifted during these years (Figures 3 and 5), with an increase in proportion of planted tree C and decrease of competing vegetation C pool over time (Figure 5). Similar results were also found at the Long-Term Soil Productivity (LTSP) study that was installed at multiple sites across California, including one at Challenge Experimental Forest [24]. In our study, NVC appeared to accumulate greater soil C than VC for the top 20 cm, which differs from what Powers et al. [14] found. They reported no significant differences between NVC and VC over a set of study sites across northern

California and southeastern Oregon, including this study when measured at age 28. The discrepancy might be due to stand developmental stages and changes of understory species composition [24,25]. Additionally, the shift here was related with plantation stand density. An interaction between overstory and understory vegetation was significant, depending on overstory stand density, which changed over stand developmental stages [11,26].

Forest stand C pools have been assumed to generally accrue as the stand ages with a relatively linear increase after disturbance followed by a saturation phase during the final stage of stand development [27]. Although we did not measure entire C pools before age 28, the planted ponderosa tree measurements at these plots provided a similar trend, as shown by other studies [16]. The carbon accumulation rates on our plots in the vegetation and forest floor show a higher rate during the first 28 years and a decline over the next interval to age 54 (Table 4). In the earlier plantation establishment stage, it was expected that soil C would be at its lowest due to soil disturbance from site preparation following the clear cut of the previous 70-year-old natural stand. The forest floor and other aboveground C pools (i.e., planted trees and other vegetation) were also expected to be at the lowest levels at this point of stand development. After trees were planted and other vegetation were regenerated, C pools of both planted trees and other vegetation built up rapidly (and linearly) for both NVC and VC on this rather productive site [8,9,16]. Other studies have reported a buildup in the soil C pool during plantation development [28,29]. As the canopy closed, and as aboveground planted trees and other vegetation started to compete for light, water, and nutrients [30] The C pools shifted in the NVC. Zhang et al. [24] reported a larger C pool on understory vegetation than planted trees at age 5, similar C between the two pools at age 10, and reversed the trend at age 20 in the California LTSP study. The plots here should have showed similar results, although some of the lower density plots with wider spacing would not develop as rapidly as the LTSP plots where all of the trees were planted at 2.4 by 2.4 m spacing. With the exception of some shade tolerant hardwoods, further development of the stand resulted in a decrease of competing vegetation, which occurred earlier in the high density plots than in the lower density plots [11]. Our data showed a similar trend (Figures 2–5). It was slightly surprising that soil C still increased in most plots, which may be related to the addition of litter to the forest floor. The stand is now over 54 years old. As it continues to age, we expect the C sequestration rates to decline in both vegetation and soil and that the C pools will tend to reach equilibrium [31].

**Table 4.** C accumulation rates (Mg C ha$^{-1}$ yr$^{-1}$) at age 28 and 54 in the aboveground vegetation plus forest floor detritus for ponderosa pine plantations grown under three levels of planting densities and two understory vegetation control treatments.

| Age | Understory Treatment | Overstory Planting Spacings | | | |
|---|---|---|---|---|---|
| | | 1.8 × 1.8 m | 3.7 × 3.7 m | 5.5 × 5.5 m | Means |
| **28** | VC | 2.7 | 2.8 | 2.5 | 2.7 |
| | NVC | 3.1 | 3.9 | 4.1 | 3.7 |
| | **Means** | **2.9** | **3.4** | **3.3** | **3.2** |
| **54** | VC | 1.8 | 2.1 | 1.7 | 1.9 |
| | NVC | 1.4 | 0.8 | 1.0 | 1.1 |
| | **Means** | **1.6** | **1.5** | **1.4** | **1.5** |

VC = vegetation control; NVC = no vegetation control.

While the forest floor C pools in this study are comparable with other similar ponderosa pine or mixed conifer plantations in California at similar stand age, spacings, and site quality [28,32], the woody debris was significantly smaller (Tables 2 and 3). This is because all of the dead trees (standing or down) were included in the planted tree C pool. However, these numbers are much larger than forest floor C on plantations that have recently been thinned or treated with other silvicultural operations [33,34]. These thinning operations disturb the forest floor when thinned trees are dragged out of the plantations. A mountain of literature has been reported related to fuel reduction treatments

that usually include prescribed fire. The short-term effect of these treatments on forest floor was even lower [35] and would not be a fair comparison with data from the current study.

In order to address the questions about how early competing vegetation control influences the C dynamics and what stand density provides the maximum C sequestration over a long term, our answers are inconclusive, because a 54-year ponderosa pine plantation is still regarded as a young growth forest [36]. Our lower density plots have not reached the self-thinning stage. However, our results indicate that, by controlling competing vegetation early, we would expectedly enhance tree growth, which may sacrifice total C sequestration by reducing competing vegetation. Therefore, the total ecosystem C would be lower in VC than NVC treatments at age 28 (Figure 3) although it was more similar at age 54 (Figure 5). The 20-year results from LTSP California installations seemed to have the same trend [24]. The density effect (planting spacings) was not significant for C sequestration, because of the offsets among different stems and between planted pine trees and competing vegetation [31]. If no other stand conditions were considered, there is no obvious stand density that could be selected to provide the maximum ecosystem C sequestration.

Ecosystem C buildup has to depend on carbon uptake by green leaves, regardless of competing vegetation control or density manipulation. Therefore, leaf area might be a starting point to explain these results. The highest density (1.8 m $\times$ 18 m) plots include about 3000 trees ha$^{-1}$, whereas the intermediate density (3.7 m $\times$ 3.7 m) has about 750 trees ha$^{-1}$ and the lowest density (5.5 m $\times$ 5.5 m) has 330 trees ha$^{-1}$. So the highest density plots in VC would have more leaf area during the earlier stand developmental years, which should promote greater pine growth and greater pine litterfall. The other density plots should slowly catch up and perhaps surpass, once subplots at all spacings reach a stable leaf area index. Because the gross primary production is spread over many smaller trees, the tree growth efficiency (biomass accumulation per leaf area) in the highest density plots decreased with stand development [24]. Similar trends have been reported in ponderosa pine grown in southeastern Oregon [37]. If this reasoning is true, then the highest C accumulation in the intermediate density (3.7 m) at age 28 suggest that vegetation at 1.8 m spacing had been stressed and physiologically less efficient, but vegetation at 5.5 m spacing had not reached stable leaf area index (Figure 3). By year 54, the lowest density plots had caught up and the density effect seemed to have disappeared in the VC treatment (Figure 5).

The NVC plot dynamics is relatively complicated, because of the interaction between the competing vegetation and overstory trees [24]. Presumably, we might assume that all NVC spacings have the same leaf area as the entire subplot area would have been occupied by vegetation early on, regardless of planted trees or competing species [26]. As the stands became older, the lowest density plots may have developed even more leaf area, as it had larger trees that carry more needle mass as well as more live, competing vegetation including hardwood species with high leaf area (Figure 6). Under this scenario, greater C accumulation on the NVC subplots, especially at lower density subplots, would be due to an overall greater leaf area.

Finally, it is essential to consider the effects of wildfires and biotic disturbances, such as *Dendroctonus* bark beetles, on pine dominant ecosystems when managing forests for carbon. Abundant evidence demonstrates that in general, lower density stands are more fire resistant and less susceptible to bark beetle infestation [3,38,39]. Although we did not measure the probability of wildfire severity and possible beetle damage in these subplots, larger pine trees with thicker bark with a higher crown base would seem to be more fire resistant than smaller trees with a lower crown base [38,39]. Therefore, our consistently smaller trees in the NVC or higher density subplots would likely suffer much heavier mortality than the larger trees in the VC or lower density subplots [4,39,40]. In addition, the greater amounts of woody debris in the NVC subplots from competing vegetation, especially from dead or dying shrubs that result from overstory shading, would become surface fuels, increasing the risk of wildfires. Finally, we cannot overlook the benefits of lower density or thinning stands for growing larger trees faster by ameliorating tree water or nutrient stresses. It is well established from a variety of studies that lowering stand density would not sacrifice productivity in ponderosa pine

stand [4,11,31,41]. Unlike many smaller-sized trees and understory shrubs, large trees in our lower density and/or the VC would be converted to forest products that store C much longer [42]. While the smaller-sized trees in the high density or the NVC subplots showed similar amounts of C as subplots with larger trees, these smaller trees are usually not useable for products and, therefore, not harvested, with the C storage likely ultimately being lost due to disturbance-related mortality.

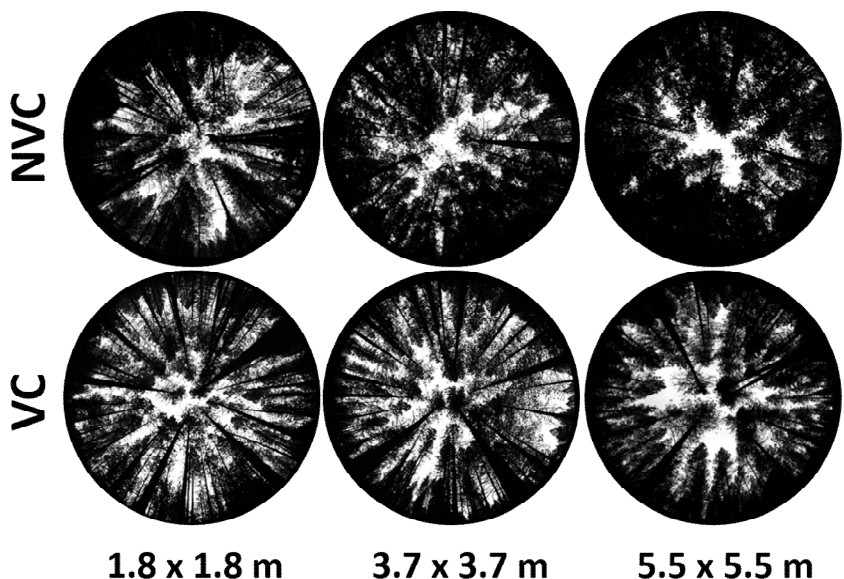

**Figure 6.** Example fisheye photos taken at age 54 from the center of each subplot under combinations of understory vegetation control (VC) and non-control (NVC) and three different levels of planting spacings at Challenge Experimental Forest, California, USA.

## 5. Conclusions

Five major carbon pools of this ponderosa pine plantation measured at ages 28 and 54, demonstrated that planted stand density did not affect the total ecosystem C, but shifted carbon pools as the stand developed. The NVC appears to accumulate greater C than the VC in the early years because of competing vegetation ingrowth. By age 54, the differences between the treatments narrow as the pines continue to grow larger in the VC, which offsets the effect of the ingrowth in NVC. Simultaneously, other competing vegetation C was reduced by age 54, presumably due to shading from the planted overstory pines. The detritus is not significantly different among treatments in either measurement year. While NVC seems to accumulate more C early on, the differences from the VC were rather subtle. Clearly, as the stands continue to grow, the total C of the larger pines of the VC may overtake the total C of the NVC. Taking into account the need for forest resiliency to frequent wildfires and bark beetles in the western United States or across the world, we conclude that we must pay attention to promoting growth of overstory trees by early controlling the understory vegetation to manage forests for long-term carbon storage.

**Supplementary Materials:** The following are available online at http://www.mdpi.com/1999-4907/11/9/997/s1, Table S1: Analyses of variance for testing treatment effects on carbon pools in a ponderosa pine plantation grown in northern California at age 28 and 54. Table S2: Probability (Pr > F) for testing treatment effect with PROC MIXED in SAS for soil carbon at age 28 and 54.

**Author Contributions:** Conceptualization, J.Z. (Jianwei Zhang) and J.Z. (Jie Zhang); methodology, J.Z. (Jianwei Zhang) and K.M.; formal analysis, J.Z. (Jie Zhang) and J.Z. (Jianwei Zhang); investigation, J.Z. (Jianwei Zhang), K.F., K.M., and J.Z. (Jie Zhang); writing—original draft preparation, J.Z. (Jie Zhang) and J.Z. (Jianwei Zhang); writing—review and editing, J.Z. (Jie Zhang), J.Z. (Jianwei Zhang), K.F., and K.M.; visualization, K.F. and J.Z. (Jianwei Zhang); supervision, J.Z. (Jianwei Zhang). All authors have read and agreed to the published version of the manuscript.

**Funding:** This research received no external funding except for US Forest Service, Pacific Southwest Research Station. As a visiting scholar here, Jie Zhang was supported by a scholarship from China Scholarship Council (No. 201706600025).

**Acknowledgments:** We sincerely thank all people who were involved any activities during various stages of this long-term study. We especially thank William Oliver who designed and oversaw the installation and early data collection and late Robert Powers who oversaw the age 28 measurements. The comments from two anonymous reviewers improved the manuscript are greatly appreciated. Use of trade names in this paper does not constitute endorsement by the United States Forest Service.

**Conflicts of Interest:** The authors declare no conflict of interest.

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
