# Peer review of "Effect of Silviculture on Carbon Pools during Development of a Ponderosa Pine Plantation"

_forests, doi:10.3390/f11090997_

Round 1
Reviewer 1 Report
Presented paper deals with comparison of carbon pools in pondersosa pine plantation in California (USA) between 26 years. More carbon pools are analyzed: live planted trees, dead planted trees, live competing vegetation, coarse woody debris, O horizon and Soil from depth 0-20 cm. Those carbon pools represents by my opinion relevant amounts of whole carbon in forest ecosystems. Comparison of Carbon amounts by development in really planted identical forest inclusive Carbon in background I find as very interesting for current research of Carbon sequestration and very important for decisions of next management.
I have very satisfied with presented study from succinct abstract, well-combined Introduction, appropriated chosen design described in Material and Methods, transparent and suitable Results to briefly Discussion.
I have to paper followed little comments and recommendations:
- Fig. 1. Detailed spacing study Plot is hard readable, here too absent scaling factor. I recommend expand share of this picture and too change border color of Experiments (I can´t see 10 plots). Sign, please, controlled and no controlled subplots. Maybe, background ortophoto is not needful for better view. You can little reduce share of Map of California and too Experimental forest.
- Specify shape and area of experimental plots, I believe, that Plots are different.
- Author presents two different ages: 28 and 54. Experiment was establish in 1966. How old were pine seedling? One or two years? Please, append too years of realized measurements – maybe in years 1982 and 2018?
- Why were changed by competing vegetation area and shape of plots from circular 0,785 m2 to square 0,25 m2? It was reduced to 1/3, I consider this fact as unsuitable.
- In Results (row 199) authors mentioned Table S1, this table absent in presented text.
- Please, improve quality of Fig 2 , 3 and 4.
- In table 1 and table 2 append values of variability to each result, I recommend either standard deviation or standard error, append mark ± after mean value
- Explain evaluation of Mortality (Fig. 4) in Methods. It is Carbon amount of presented standing or lying dead trees in age 54? Or? By competing vegetation you can´t find dead trees?
Author Response
Responses to Reviewer 1:
Review 1: Presented paper deals with comparison of carbon pools in pondersosa pine plantation in California (USA) between 26 years. More carbon pools are analyzed: live planted trees, dead planted trees, live competing vegetation, coarse woody debris, O horizon and Soil from depth 0-20 cm. Those carbon pools represents by my opinion relevant amounts of whole carbon in forest ecosystems. Comparison of Carbon amounts by development in really planted identical forest inclusive Carbon in background I find as very interesting for current research of Carbon sequestration and very important for decisions of next management.
I have very satisfied with presented study from succinct abstract, well-combined Introduction, appropriated chosen design described in Material and Methods, transparent and suitable Results to briefly Discussion.
Thank you for your satisfaction of our story here. In fact, a long-term research infrastructure such as this one is rare and I feel very fortunate inheriting many long-term research plots.
I have to paper followed little comments and recommendations:
- Fig. 1. Detailed spacing study Plot is hard readable, here too absent scaling factor. I recommend expand share of this picture and too change border color of Experiments (I can´t see 10 plots). Sign, please, controlled and no controlled subplots. Maybe, background ortophoto is not needful for better view. You can little reduce share of Map of California and too Experimental forest.
I am sorry that the system provided you a low pixel pdf. I hope that final publication will be as clear as in my word file as well as pdf file that I made. All scales are in their respective map and they are readable. Therefore, we decide to leave as it is.
- Specify shape and area of experimental plots, I believe, that Plots are different.
They are clearly rectangular plots and square subplots should this figure is in high quality. To place all plots on relatively uniform soil or land over 50 plus year ago, previous scientists came up this idea that each subplot (or plot) included the same numbers of trees. Therefore, plot size varies. I clarify this in the revised text.
- Author presents two different ages: 28 and 54. Experiment was establish in 1966. How old were pine seedling? One or two years? Please, append too years of realized measurements – maybe in years 1982 and 2018?
We provide these details with one-year-old seedlings were planted. Two sampling years are in 1993 and 2019.
- Why were changed by competing vegetation area and shape of plots from circular 0,785 m2 to square 0,25 m2? It was reduced to 1/3, I consider this fact as unsuitable.
At age 28, all understory vegetation including hardwood species and big shrubs were collected within these circular plots provided these species were luxuriant. Whereas at age 54, most shrubs were dead and were samples as forest floor or coarse wood debris, and hardwood grew big and we estimated their separately using the allometric equations. So, smaller sampling areas would be sufficient to capture herbaceous species and a few small stature shrubs.
- In Results (row 199) authors mentioned Table S1, this table absent in presented text.
It is in the supplementary file – should be separately from the main text. Reviewer 2 made some comments, indicating that it was reviewed. If reviewer 2 was able to access it, I assume that you should be too.
- Please, improve quality of Fig 2 , 3 and 4.
These are really good quality, which suggests that the journal editorial staff should have consider to provide reviewers a quality of manuscript for review.
- In table 1 and table 2 append values of variability to each result, I recommend either standard deviation or standard error, append mark ± after mean value
This study replicated only twice. Some standard deviations are relative big, which weakens our inferences. Let’s go without them.
- Explain evaluation of Mortality (Fig. 4) in Methods. It is Carbon amount of presented standing or lying dead trees in age 54? Or? By competing vegetation you can´t find dead trees?
I do not know how to explain because a dead tree is a dead tree. The same allometric equation was used to estimate the biomass. Dead tree C was clearly shown in the figure.
Reviewer 2 Report
General statement:
Manuscript is properly written, however, in few places should be redrafted. For instance, the Authors try to explain some results in the results section (see my detailed comments below) what is not a proper place for it. The language is properly used, however, sometimes it was quite difficult for me to feel the flow (especially in the discussion section). I have only few thin remarks about language used, however, as I am not an English native speaker, it could be connected my incomprehension (details below).
Abstract could better brief the manuscript content. The second phrase is very important however it does not reflects the content. Nevertheless, I would recommend to leave this sentence and add some information about the influence of thinning on the soils and carbon balance. I suggest to add it in the introduction section (with proper references) but also to mention it in conclusion. Moreover, I propose to add to the abstract the main values obtained in the study and to underline that even if total carbon budget in the ecosystems is similar between treatments, the share of particular compartments of this budget changes.
Introduction is well (briefly) written, however there is a lack of short information about the advantage of multilayered stands in total carbon budget (see my comment to the abstract), which enable to avoid clearcuttings and makes the soil “untouched” by management activity.
Materials and methods should bear some more sufficient information. For details see my remarks below.
Results section is mostly clearly written. I have only few remarks (see detail below). Beside this I propose to anchor figures 2 and 4 (and 3 and 5) close to each other. It could improve clarity, however I leave it under the Authors decision.
Discussion section is arranged in a good way. Study statements are mainly based good enough on cited literature, however the last part should be developed (see my comments below).
Detailed remarks (The Authors could find them also in pdf file):
L32: Not only there.
L60: add hyphen
L68: I propose to use either year or yr but in all hyphenated cases after first mention.
L101: Could it be possible that this herbicide treatment influenced also growth of ponderosa pine at first stage of living? Is it possible to add one phrase in that matter?
L109-111: I propose to substitute this phrase by "...at the ponderosa pine age of 28 and 54 when..." or something like that as not all information are necessary.
L111-112: This is not a necessary information. I propose to delete it.
L114: Were the tree spacing there same as within study plots?
L131-132: It would be good if you show these results briefly.
L135: In my opinion it is disadvantage of the study. The biomass allocation patterns are much different for pines at age 54 compared to these at age 28. It could give you relatively high over- or underestimation for particular tree components. If you did not fell the trees at the age of 54, please consider to use some local-specific allometric equations from literature. Are there any? Compare the results. Maybe the use such equations will be more justified.
L137: It's better to display ratios as a fraction. So change it please into 0.22.
L141-142: Please precise it if you mean other tree species. If you don't mean so it is inconsistent with previous phrase.
L145: Is this correct? It seems that these plants grew in lab before. I could be wrong as I am not an English native speaker.
L147: Five per each subplot? Or per NVC subplots only? Or five per all plots/subplots? Please precise it.
L148: ...of at least XXX g. Please precise it.
L150: cancel italics please.
L154: Please use a fraction. 0.25
L170: Is that correst? Maybe better use: "At stand age of 28,..."
L199: Actually I can't see that in Table S1.
L212: Why do you use once 1SD (Fig.2) and another time 1SE (Fig.3). I propose to unify it. If you use single SE (or SD) it would better look when you take +SE (SD).
L234-237: It sounds more like discussion. Please, move it.
L240: Actually, there are some. Do you mean statistically significant differences? Please, precise it.
L246: Why do you use once 1SD (Fig.2) and another time 1SE (Fig.3). I propose to unify it. If you use single SE (or SD) it would better look when you take +SE (SD).
L256: as above
L261 (and others yellow colored without comment): at age? Is your notation correct? Sorry for that question, but it seems that you describe process in year 28 (like 1928). I could be wrong as I am not an English native speaker.
L269-270: Please move it to discussion.
L271-272: as above
L277-279: as above.
L297: VC?
L305: in comparison to the age of 28? Please use this reference in comparisons.
L386: However, referenced paper seems to concern only ponderosa pine, doesn't it? Please, check it and if necessary redraft this phrase.
L391-397: I think this paragraph should be developed with reference to few more papers.
L402: In reference to which characteristic? Height? Volume? It would be good if you will add some table (into the methods or at the begining of results) with stand characteristics showing the diffrences in mean tree DBH, heigt, basal area etc. between stands of various spacing.
You should also mention about the understory vegetation a bit more detailed.
L403-404: This phrase, and whole paragraph is some kind of deliberation which is indirectly connected with your research. Obviously, the last paragraph of discussion is a proper place to place it. Nevertheless, as such kind of paragraphs are very open, this openness could bear some traps. Why did you mention about bark beetles outbreak or wildfires? I guess you did so to justify some of forest management activity in general carbon balance. If so, you should discuss it wider, what in turn could be a trap. Maybe it is true that lower-dense forests are more resistant to wildfires and outbreaks. However they could produce more biomass that is accumulated in leaves and branches (especially at the beginning stage of growth), which usually stays in the stands after cleanings (that are a carbon source when mineralized). Denser stands could produce more biomass in well-quality trunks, which are used to produce some more long-lasting goods, what ties carbon for longer time than decaying wood debris in the forest floor.
By this example I just wanted you to try to discuss about trees spacing pros and cons wider. Anyway, table with detailed stand characteristic would be very helpfull to imagine what kind of stands are you comparing in this paper (see my previous comment).
L419: You could also mention that multi-layered forests enables to avoid clearcuttings, what in turn let the soil to stay covered by vegetation. Otherwise, the soil (mainly O horizon) is a carbon source for many years after clearcuttings.
L421: Please explain, what CWD means. Explain also different collours of values.

Author Response
Please see the attached responses.
Thank you for your excellent reviews!
Responses to Reviewer 2: Comments and Suggestions for Authors
Thank you very much for spending time on this manuscript. It is a very good review. We have tried to address all the comments. Clearly, these changes have significantly improved the manuscript. I will address them line by line later.
General statement:
Manuscript is properly written, however, in few places should be redrafted. For instance, the Authors try to explain some results in the results section (see my detailed comments below) what is not a proper place for it. The language is properly used, however, sometimes it was quite difficult for me to feel the flow (especially in the discussion section). I have only few thin remarks about language used, however, as I am not an English native speaker, it could be connected my incomprehension (details below).
Abstract could better brief the manuscript content. The second phrase is very important however it does not reflects the content. Nevertheless, I would recommend to leave this sentence and add some information about the influence of thinning on the soils and carbon balance. I suggest to add it in the introduction section (with proper references) but also to mention it in conclusion. Moreover, I propose to add to the abstract the main values obtained in the study and to underline that even if total carbon budget in the ecosystems is similar between treatments, the share of particular compartments of this budget changes.
Introduction is well (briefly) written, however there is a lack of short information about the advantage of multilayered stands in total carbon budget (see my comment to the abstract), which enable to avoid clearcuttings and makes the soil “untouched” by management activity.
This is a good idea but not appropriate for this study because the second story naturally regenerated from both hardwoods and shade tolerant conifers have formed in these plots (subplots) regardless stand density. It is meaningless to compare these treatments. Therefore, we did not mention it.
Materials and methods should bear some more sufficient information. For details see my remarks below.
Results section is mostly clearly written. I have only few remarks (see detail below). Beside this I propose to anchor figures 2 and 4 (and 3 and 5) close to each other. It could improve clarity, however I leave it under the Authors decision.
We left where they were. After final version, I hope that the manuscript will be tightly arranged, which may improve the readability.
Discussion section is arranged in a good way. Study statements are mainly based good enough on cited literature, however the last part should be developed (see my comments below).
The last paragraph has been considerably revised by following your comments.
Detailed remarks (The Authors could find them also in pdf file):
L32: Not only there.
The sentence has been rewritten to “We conclude that to manage forests for carbon, we must pay more attention to promoting growth of overstory trees by early controlling competing vegetation, which will provide more opportunities for foresters to create resilient forests to disturbances and store C longer in a changing climate.”
L60: add hyphen
L68: I propose to use either year or yr but in all hyphenated cases after first mention.
All are changed to “year”.
L101: Could it be possible that this herbicide treatment influenced also growth of ponderosa pine at first stage of living? Is it possible to add one phrase in that matter?
Yes, we clarify this as “On one subplot, competing vegetation was controlled by 2,4,5-T (2,4,5-trichlorophenoxy acetic acid) applied by hand sprayer in the second and fourth years after planting. Subsequent competing woody plants were removed by hand for about five more years. The herbicide spray drops were never touched on planted seedlings and therefore should not affect their performance.”
L109-111: I propose to substitute this phrase by "...at the ponderosa pine age of 28 and 54 when..." or something like that as not all information are necessary.
Changed as recommended.
L111-112: This is not a necessary information. I propose to delete it.
Deleted.
L114: Were the tree spacing there same as within study plots?
Yes, they are, we clarify it by adding a sentence in section 2.1. Study site and original design
L131-132: It would be good if you show these results briefly.
We were thought about that and decided not to include them because those information would overshadow our main objectives in the paper. In addition, if you had provided stem biomass, people would like to know branches or leaves. Then, our C story drifts to develop the allometric equations.
L135: In my opinion it is disadvantage of the study. The biomass allocation patterns are much different for pines at age 54 compared to these at age 28. It could give you relatively high over- or underestimation for particular tree components. If you did not fell the trees at the age of 54, please consider to use some local-specific allometric equations from literature. Are there any? Compare the results. Maybe the use such equations will be more justified.
Excellent thought! Although ponderosa pine is one of the most studied species in the West and many people may have done some individual tree biomass to fit an allometric equation, this Forest Service laboratory had collected a considerable amount of biomass from 10-year-old trees to 250-year-old. I have published one for a similar purpose of this paper over ten years ago. I did calculate aboveground biomass with previous one and compared with the current one for these trees (dbh classes), and results were very similar. I decided to use the current one, which was developed at the site. You are right, when trees grow, they change allocation patterns. Yet, I find that total AGB does not change much, which is another reason that I did not separately estimate leaf, branch, and stem biomass.
L137: It's better to display ratios as a fraction. So change it please into 0.22.
Great, I changed 22% to 0.22.
L141-142: Please precise it if you mean other tree species. If you don't mean so it is inconsistent with previous phrase.
The sentence has been revised.
L145: Is this correct? It seems that these plants grew in lab before. I could be wrong as I am not an English native speaker.
Changed to “samples were brought to the lab”.
L147: Five per each subplot? Or per NVC subplots only? Or five per all plots/subplots? Please precise it.
Five sampling spots per subplot; we clarified it.
L148: ...of at least XXX g. Please precise it.
We added the gram.
L150: cancel italics please.
We did.
L154: Please use a fraction. 0.25
The 0.25 was used.
L170: Is that correst? Maybe better use: "At stand age of 28,..."
Good suggestion – later one all year 28 or year 54 have been changed to age 28 and age 54.
L199: Actually I can't see that in Table S1.
It is there. This is a philosophical argument. Probability is never an exact, in this case, it shows 0.036. I always think that P value is always “<” or “>”. So, I used P < 0.04.
L212: Why do you use once 1SD (Fig.2) and another time 1SE (Fig.3). I propose to unify it. If you use single SE (or SD) it would better look when you take +SE (SD).
Good eyes, it should have been SD in the figure. I made an error. It has been corrected and so has 1 issue.
L234-237: It sounds more like discussion. Please, move it.
The speculations were deleted.
L240: Actually, there are some. Do you mean statistically significant differences? Please, precise it.
You are right, a significant was added before difference.
L246: Why do you use once 1SD (Fig.2) and another time 1SE (Fig.3). I propose to unify it. If you use single SE (or SD) it would better look when you take +SE (SD).
Same as above; it should have been SD in the figure. It has been corrected and so has 1 issue.
L256: as above
We kept it because there is no specific discussion paragraph on it. But, readers want to know the reasons or perhaps we want to know. We can delete them if you insist to do so.
L261 (and others yellow colored without comment): at age? Is your notation correct? Sorry for that question, but it seems that you describe process in year 28 (like 1928). I could be wrong as I am not an English native speaker.
Good catch, as I stated earlier, year # has changed to age #.
L269-270: Please move it to discussion.
L271-272: as above
L277-279: as above.
The same reasons to keep them. We can delete them if you insist to do so.
L297: VC?
It should have been VC, Thanks.
L305: in comparison to the age of 28? Please use this reference in comparisons.
Yes, we did in the revised text.
L386: However, referenced paper seems to concern only ponderosa pine, doesn't it? Please, check it and if necessary redraft this phrase.
Good catch – the sentence has changed to: “Similar trends have been reported in ponderosa pine grown on southeastern Oregon”.
L391-397: I think this paragraph should be developed with reference to few more papers.
We tried and added some references and photos as Figure 6 to make the point. If this is still not satisfied your and editor’s requirement, we can delete this paragraph.
L402: In reference to which characteristic? Height? Volume? It would be good if you will add some table (into the methods or at the begining of results) with stand characteristics showing the diffrences in mean tree DBH, heigt, basal area etc. between stands of various spacing.
You should also mention about the understory vegetation a bit more detailed.
Yes, it is a good idea to add such table. We have added it as a Table 1 in the revised manuscript.
L403-404: This phrase, and whole paragraph is some kind of deliberation which is indirectly connected with your research. Obviously, the last paragraph of discussion is a proper place to place it. Nevertheless, as such kind of paragraphs are very open, this openness could bear some traps. Why did you mention about bark beetles outbreak or wildfires? I guess you did so to justify some of forest management activity in general carbon balance. If so, you should discuss it wider, what in turn could be a trap. Maybe it is true that lower-dense forests are more resistant to wildfires and outbreaks. However they could produce more biomass that is accumulated in leaves and branches (especially at the beginning stage of growth), which usually stays in the stands after cleanings (that are a carbon source when mineralized). Denser stands could produce more biomass in well-quality trunks, which are used to produce some more long-lasting goods, what ties carbon for longer time than decaying wood debris in the forest floor.
By this example I just wanted you to try to discuss about trees spacing pros and cons wider. Anyway, table with detailed stand characteristic would be very helpfull to imagine what kind of stands are you comparing in this paper (see my previous comment).
These are good suggestions and we have tried to incorporate these ideas to the discussion with appropriate references. The entire paragraph has been rewritten.
L419: You could also mention that multi-layered forests enables to avoid clearcuttings, what in turn let the soil to stay covered by vegetation. Otherwise, the soil (mainly O horizon) is a carbon source for many years after clearcuttings.
As I indicated earlier, this is a good idea but not appropriate for this study because the second story naturally regenerated from both hardwoods and shade tolerant conifers have formed in these plots (subplots) regardless stand density and vegetation control. There are a plenty of ingrowing shade tolerant trees (not much shrubs) grown even in high density plots due to the mortality–caused gaps and in the VC subplots after a cease of vegetation control. So, it makes comparisons not interesting. Therefore, we did not mention it.
L421: Please explain, what CWD means. Explain also different collours of values.
We have changed all CWD to woody debris throughout the manuscript. There should not include any CWD (coarse woody debris) any more. Explanations for color highlights and bold text were provided.
Round 2
Reviewer 2 Report
Dear Authors, Dear Editor
The manuscript is improved much enough according to my suggestions. The Authors did not agree with some of them however they were not crucial for improving the paper. I have only few comments which should not suspend the publication. They result likely from some misunderstanding. I have recommended to add some information about the influence of thinning on the soils and carbon balance. What I meant by "thinning" was to discuss in the introduction and conclusion sections the differences in carbon balance between stands with controlled and non-controlled vegetation, where some of the species regenerating under ponderosa pine canopy could (as I believe) recruit in the future to the main canopy and substitute the present one. This natural process enable to keep soil relatively untouched compared with soil preparation before planting within the stands without undergrowth. I did not mean to "compare" the stands of various spacing in that matter. This changes could in my opinion improve the story. Nevertheless, I will not insist on this.
Similarly the second thing. Some part of results are spiced up with phrases that are more proper for discussion. I prefer to divide results and discussion, however if Editors decide to leave it at it is, I will not stand against it.
The third remark concerns the methods used. I have marked them in line 135 (of first version), however I should do that for lines 113-115. I do not understand why the Authors estimates the biomass of tree set based only on equatation developed on measurements made only for trees at age 28. Especially that as it was written, the Authors would have some equations that are more proper for trees at age 54.